# An Interpretable Machine Learning Framework for Analyzing the Interaction Between Cardiorespiratory Diseases and Meteo-Pollutant Sensor Data

**DOI:** 10.3390/s25154864

**Published:** 2025-08-07

**Authors:** Vito Telesca, Maríca Rondinone

**Affiliations:** Department of Engineering, University of Basilicata, 85100 Potenza, Italy; vito.telesca@unibas.it

**Keywords:** cardiorespiratory diseases, air pollution, interpretable machine learning, benchmarking strategy, SHAP and LIME analysis

## Abstract

This study presents an approach based on machine learning (ML) techniques to analyze the relationship between emergency room (ER) admissions for cardiorespiratory diseases (CRDs) and environmental factors. The aim of this study is the development and verification of an interpretable machine learning framework applied to environmental and health data to assess the relationship between environmental factors and daily emergency room admissions for cardiorespiratory diseases. The model’s predictive accuracy was evaluated by comparing simulated values with observed historical data, thereby identifying the most influential environmental variables and critical exposure thresholds. This approach supports public health surveillance and healthcare resource management optimization. The health and environmental data, collected through meteorological sensors and air quality monitoring stations, cover eleven years (2013–2023), including meteorological conditions and atmospheric pollutants. Four ML models were compared, with XGBoost showing the best predictive performance (R^2^ = 0.901; MAE = 0.047). A 10-fold cross-validation was applied to improve reliability. Global model interpretability was assessed using SHAP, which highlighted that high levels of carbon monoxide and relative humidity, low atmospheric pressure, and mild temperatures are associated with an increase in CRD cases. The local analysis was further refined using LIME, whose application—followed by experimental verification—allowed for the identification of the critical thresholds beyond which a significant increase in the risk of hospital admission (above the 95th percentile) was observed: CO > 0.84 mg/m^3^, P_atm ≤ 1006.81 hPa, Tavg ≤ 17.19 °C, and RH > 70.33%. The findings emphasize the potential of interpretable ML models as tools for both epidemiological analysis and prevention support, offering a valuable framework for integrating environmental surveillance with healthcare planning.

## 1. Introduction

In recent years, the growing impact of cardiorespiratory diseases on public health has renewed interest in analyzing environmental factors that can influence their onset and exacerbation [1]. Diseases involving both the respiratory and cardiovascular systems—such as chronic obstructive pulmonary diseases combined with heart failure or respiratory flare-ups in patients with prior ischemic events—represent a critical area from an epidemiological standpoint, not only due to their prevalence but also because of their strong seasonality and sensitivity to external variables [2].

Numerous studies have shown that exposure to air pollutants—especially fine particulate matter (PM_2.5_ and PM_10_), nitrogen dioxide (NO_2_), carbon monoxide (CO), and ground-level ozone (O_3_)—can trigger systemic inflammatory responses and physiological imbalances affecting both respiratory function and cardiovascular health [3]. Additionally, meteorological conditions such as low temperatures, high humidity, and unstable atmospheric pressure can create environments conducive to the exacerbation of chronic illnesses and the occurrence of acute episodes [4]. In densely populated urban settings, the combination of these variables may have a significant health impact, particularly on vulnerable populations—elderly people, children, and individuals with comorbidities—who are continuously exposed to physically stressful environments [5].

Recent research has also emphasized the interaction between meteorological parameters and air pollution, showing that extreme temperatures or humidity can significantly amplify the adverse effects of pollutants on cardiovascular outcomes. For instance, short-term exposure to PM_2.5_ during low-temperature or high-humidity days has been associated with an increased risk of cardiovascular admissions [6].

In recent years, artificial intelligence (AI) and machine learning (ML) techniques have shown promise in environmental and healthcare applications, particularly in forecasting acute events linked to environmental exposure. Algorithms such as Random Forest, XGBoost, and deep neural networks have been adopted to model the impact of meteorological and pollutant variables on hospital admissions, emergency visits, and mortality [7,8,9].

Several recent studies have demonstrated the potential of ML models—including ensemble approaches and time-series deep learning techniques—in predicting hospital admissions for both respiratory [10,11] and cardiovascular conditions [12,13]. 

These models vary in complexity and interpretability, ranging from linear regressors to ensemble methods like stacking [13], with performance often depending on the data resolution, feature selection methods, and forecast horizon.

However, despite these advances, many existing models lack sufficient interpretability and generalizability, limiting their applicability in clinical and environmental health decision-making. Moreover, prior studies often do not fully integrate high-resolution environmental data with comprehensive interpretability analyses, leaving a gap that this research aims to fill.

The motivation behind this study is to address these limitations by developing and benchmarking machine learning models that balance predictive accuracy with interpretability, leveraging extensive environmental and hospital admission datasets.

In parallel, comprehensive reviews have also emphasized the cumulative burden of environmental stressors—including air pollution, noise, artificial light, and climate extremes—on cardiovascular disease progression and outcomes, urging integration of these variables in predictive modeling [14].

A key aspect in such studies is the quality and reliability of environmental data. The precision of measurements—obtained through certified sensor networks distributed across the territory—directly influences the performance of AI models. In particular, the availability of high-resolution temporal and spatial data enables a more accurate representation of environmental exposure conditions, enhancing the model’s sensitivity to local variations [15,16,17].

The present study focuses on the metropolitan area of Bari, a Mediterranean city located in Southern Italy, characterized by a temperate climate, variable air quality conditions, and a high seasonal burden on emergency care services.

This study shows an analysis based on machine learning techniques, aimed at investigating the relationships between environmental parameters—collected through distributed meteorological and environmental sensors managed by official monitoring networks—and daily emergency department admissions due to cardiorespiratory conditions. The modeling follows a benchmarking strategy [18] in which four regression models—Random Forest, XGBoost, LightGBM, and Explainable Boosting Machine (EBM) [19,20,21,22,23]—are trained and compared in terms of test performance to determine the best-performing model.

All algorithms were optimized using the Optuna library [24], which enables an efficient hyperparameter search through Bayesian optimization and early stopping techniques [25]. This approach enables the tuning of model performance based on the specific characteristics of the case study dataset, composed of over ten years of daily environmental and hospitalization data. Model effectiveness was evaluated using a set of quantitative metrics—including mean squared error, mean absolute error, and the coefficient of determination R^2^ [26]—as well as an analysis of the error distribution on the test sets.

The results show that the selected models achieve strong predictive performance, with XGBoost showing a favorable balance between accuracy and interpretability.

Beyond predictive performance, one of the central goals of this work is model transparency and interpretability. Therefore, in addition to the prediction phase, considerable attention was given to analyzing model behavior through both global and local interpretation tools [27,28,29,30]. In particular, the use of SHAP (SHapley Additive exPlanations) enabled an in-depth understanding of the variables most influencing the model outputs, both at an aggregate level (via Bee Swarm plots) and at the individual level (via SHAP dependence plots) [31]. The most influential variables—namely, carbon monoxide, atmospheric pressure, average temperature, and relative humidity—were further explored through local analysis.

These insights were deepened by integrating SHAP analysis with LIME (Local Interpretable Model-Agnostic Explanations) [32]. Following proper experimental verification, the resulting plots enabled the quantification of each predictor’s marginal effect and the identification of environmental thresholds beyond which the risk of hospitalization increases significantly.

Although the study was conducted in a specific Mediterranean context, the proposed methodological framework is general and can be applied and validated in different climatic and environmental settings.

This article is organized as follows. Section 2 describes the materials and methods, including the adopted methodology, the dataset used, the artificial intelligence models employed, and the interpretability techniques applied. Section 3 presents the results obtained, with a detailed analysis of the data and an evaluation of the global and local interpretability of the models. Section 4 is dedicated to a discussion, where the results are interpreted and compared with the existing literature, also highlighting the verification of the interpretability analyses. Finally, Section 5 reports the study’s conclusions, outlining limitations and possible future developments.

### State of the Art

In recent years, several studies have employed machine learning techniques to investigate the relationship between environmental exposure and hospitalizations due to cardiovascular and respiratory diseases. These studies have adopted various approaches in terms of predictive models, integrated environmental data (air pollutants, meteorological variables), forecast time scales, and the spatial resolution of datasets.

For instance, Ravindra et al. (2023) used Random Forest models to predict outpatient visits for acute respiratory infections, highlighting air pollution as a significant triggering factor [11].

A similar approach was adopted by Kurucz et al. (2024), who employed multiple linear regression and Random Forest to forecast emergency department presentations for acute coronary syndrome, confirming the presence of seasonal patterns and significant daily fluctuations [12].

Hu et al. (2020) proposed a stacking ensemble model to predict daily cardiovascular admissions, showing substantial improvements over individual base models [13].

Similarly, He et al. (2022) explored the interactive effects of meteorological and air pollution factors on cardiovascular hospitalizations, identifying synergistic effects between low temperatures and elevated PM concentrations [6].

Lu et al. (2021) [10] focused on short-term forecasting of emergency respiratory visits in Beijing using ARIMA, MLP, and LSTM models. Their findings emphasize the predictive strength of fine particulate matter (PM_2.5_) and demonstrate the feasibility of hybrid ML approaches [10].

Finally, the comprehensive review by Münzel et al. (2021) synthesized existing evidence on the effects of not only physical but also psychosocial environmental factors (e.g., noise, artificial light) on cardiovascular health, also proposing mitigation guidelines [14].

These contributions confirm the effectiveness of ML models in supporting health decision-making based on evolving environmental conditions, while also emphasizing the need for interpretable, validated, and locally adaptable models.

A summary of the main characteristics and findings of these studies is presented in Table 1.

The analyzed studies show significant progress in applying machine learning to predict hospital admissions related to environmental exposure. However, some limitations remain. Most works focus primarily on predictive performance, overlooking the importance of model interpretability, which is essential to support clinical and policy decisions.

Moreover, many models use datasets with limited spatial or temporal resolution, reducing their adaptability to different local contexts.

Another gap concerns the lack of systematic comparative evaluations among different machine learning algorithms under homogeneous conditions, which are necessary to identify the most suitable models for specific predictive tasks.

Finally, few studies thoroughly analyze the environmental thresholds beyond which the risk of hospitalization significantly increases, thus limiting practical guidance.

This work aims to fill these gaps by adopting a benchmarking strategy on a high-resolution dataset spanning over ten years, combining advanced interpretability techniques (such as SHAP and LIME). Additionally, it focuses on identifying critical environmental variables and risk thresholds, enhancing the practical applicability of the models in local healthcare management.

## 2. Materials and Methods

### 2.1. Methodology

The workflow adopted in this study is illustrated in Figure 1 and consists of a series of analytical and computational steps, described in detail below.

### 2.2. Dataset Collection and Preparation

The data used refer to the period 2013–2023 and include meteorological and environmental variables used to identify correlations between these and the daily number of hospital admissions for cardiorespiratory causes. After a thorough preprocessing phase, missing values in the target variable were removed, as well as any non-numeric columns, such as those containing temporal information. This removal was performed to prevent temporal information that is not directly modelable from influencing predictive performance in an uninterpretable way. The dataset was then split into a training set (70%) and a test set (30%) using a random but reproducible procedure, keeping the statistical distribution of the target unchanged.

### 2.3. Machine Learning Models

The modeling process involved training and evaluating on the test set four regression models:Random Forest (RF): A supervised learning algorithm based on an ensemble of decision trees, where each tree is trained on a random subsample of the dataset. This strategy introduces a high degree of diversity among trees, reducing variance and improving generalization. In regression, the final prediction is calculated as the mean of the individual tree predictions [33]. Random Forest is known for its robustness to noise, ability to handle nonlinearly correlated features, and lack of assumptions about data distribution. Additionally, it shows relatively stable behavior against overfitting, especially with a large number of trees [34].Extreme Gradient Boosting (XGBoost): One of the most advanced and optimized implementations of gradient boosting. The algorithm builds decision trees sequentially, where each new tree is trained to minimize the residual errors of the previous ones through gradient descent on a differentiable loss function [35]. XGBoost integrates regularization techniques (L1 and L2), efficient memory management, parallel training, and early pruning to prevent overgrowth of trees. It is particularly effective on structured and high-dimensional datasets but requires careful hyperparameter tuning for optimal performance. Its popularity also stems from the computational speed and high predictive accuracy [20].Light Gradient Boosting Machine (LightGBM): A boosting algorithm that uses decision trees as base learners and is designed to be highly efficient in computation and memory. LightGBM builds trees leaf-wise (instead of level-wise), selecting the leaf with the greatest potential loss reduction at each iteration. This approach yields deeper and more accurate models but requires careful regularization to avoid overfitting. LightGBM supports missing value handling and customized loss functions, making it especially suitable for large and heterogeneous datasets [36].Explainable Boosting Machine (EBM): Belonging to the interpretable (glassbox) model category, it is based on the architecture of Generalized Additive Models with pairwise interactions (GAMs), balancing model transparency with high predictive performance. Each variable contributes independently to the final prediction through explicit additive functions, making it particularly suited for sensitive contexts such as healthcare [37].


To further improve performance, the models were optimized using Optuna, a Bayesian optimization framework that performs intelligent hyperparameter searches. Optuna uses a “trial-and-error” logic based on Tree-structured Parzen Estimators (TPEs) to explore and exploit promising regions of the search space more efficiently than traditional random or grid searches [25]. The models were also processed via 10-fold cross-validation, increasing their robustness and reducing the overfitting risk [38]. Their test phase performances were then evaluated and compared using the following metrics [26], which provide complementary insights into predictive accuracy and error distribution:Coefficient of Determination (R^2^): This metric measures the proportion of variance in the target variable that is explained by the model. A value close to 1 indicates excellent predictive ability. As shown in Equation (1), the coefficient of determination (R^2^) is calculated as follows:(1)$$R^{2}\;=\;1\;-\;\frac{\sum_{i=1}^{n}{\left(y_{i}-\;y\;{*}_{i}\right)}^{2}}{\sum_{i=1}^{n}{\left(y_{i}-\;y^{**}\right)}^{2}}$$

where:

y_i_ = actual value;

y_i_* = predicted value;

y_i_** = mean of the actual value;

n = number of observations.


Mean Absolute Error (MAE): This metric represents the average absolute difference between predicted and actual values. It is useful for interpreting the model’s error in the same unit as the target variable. Equation (2) provides the formula for calculating the MAE:(2)$$\mathrm{MAE}=\frac{1}{n}\sum_{i=1}^{n}\left|\;y_{i}-y_{i}^{*}\right|$$


where:

|$y_{i}-\;y_{i}^{*}$| = absolute error for each prediction;

n = number of observations.


Mean Squared Error (MSE): This metric calculates the average of the squares of the errors. It penalizes larger errors more than the MAE and is sensitive to outliers. Its expression is given by Equation (3):(3)$$\mathrm{MSE}=\frac{1}{n}\sum_{i=1}^{n}{\left(\;y_{i}-y_{i}^{*}\right)}^{2}$$


where:

($y_{i}-\;y_{i}^{*}$)^2^ = squared error for each prediction;

n = number of observations.


Root Mean Squared Error (RMSE): This metric provides the error magnitude in the same unit as the target. It helps quantify how much predictions typically deviate from actual values. Equation (4) expresses how the RMSE is computed:(4)$$\mathrm{RMSE}=\sqrt{\frac{1}{n}\sum_{i=1}^{n}{\left(\;y_{i}-y_{i}^{*}\right)}^{2}}$$


where:

√ = square root of the MSE;

n = number of observations.

### 2.4. Global and Local Interpretability (XAI Methods)

To ensure the transparency of the simulation process and to understand the extent to which each environmental variable influences the results, the proposed procedure applies both global and local Explainable Artificial Intelligence (XAI) tools.

SHapley Additive exPlanations (SHAP)

An interpretative method based on game theory that assigns a well-defined quantitative contribution to each independent variable relative to the model’s final prediction. This approach identifies the most relevant features and understands whether their effect is positive or negative on the target, supporting intuitive and consistent interpretation even in complex contexts. The analysis was conducted both globally, through the calculation of average variable importance across all observations (global SHAP feature importance), and locally, examining the specific contribution of features for each prediction (SHAP dependence plot). To facilitate result interpretation, a Bee Swarm plot was used, showing simultaneously the impact of each variable on all dataset instances, allowing for identification of critical thresholds, nonlinear behaviors, and potential interactions between variables [27,28,29,30,31].

Local Interpretable Model-Agnostic Explanations (LIME)

Used to analyze the local behavior of XGBoost, LIME builds a local linear regression model around the point of interest and shows which variables influence that particular prediction. It was used to quantitatively determine critical thresholds (subsequently validated) related to the most significant features identified by SHAP feature importance analysis [32].

### 2.5. Study Area

This work focuses on the metropolitan area of Bari, the capital city of the Apulia region, which is located in southern Italy and overlooks the Adriatic Sea. With a total population of about 1.2 million inhabitants [39], Bari represents an important urban and cultural center of the region, characterized by a strong industrial presence and significant vehicular traffic, factors that influence the local air quality. The typical Mediterranean climate of the area is characterized by hot and sunny summers accompanied by mild and rainy winters [40], meteorological conditions that can interact in a complex way with environmental factors and affect the health of the population. For the analysis carried out in this study, clinical data related to hospital admissions for cardiorespiratory diseases were used, provided by the Policlinico of Bari, one of the most important and structured hospitals in the city, with a large catchment area. In parallel, environmental and climatic data were collected through local monitoring stations, which continuously and in detail record parameters such as the concentration of atmospheric pollutants (carbon monoxide, ozone, PM10 particulate matter, and nitrogen dioxide) and meteorological variables (temperature, atmospheric pressure, and relative humidity). The integration of these data sources made it possible to build a robust and representative database, forming the basis for the subsequent machine learning modeling described below.

The choice of the metropolitan area of Bari also responds to the need to explore urban contexts characterized by temperate climatic conditions and subject to significant seasonal fluctuations within a framework of increasing climatic instability related to global climate change. This is manifested not only by rising average temperatures but also by an increase in the frequency of extreme weather events such as heat waves, heavy rains, and prolonged drought periods, often accompanied by high humidity. These dynamics can amplify the impact of atmospheric pollutants such as NO_2_, PM_2.5_, and O_3_ on public health, especially in vulnerable individuals. Moreover, the availability of data spanning a period of ten years has allowed for a detailed analysis of temporal trends and associations between environmental exposure and hospital admission incidence. This approach enables us to capture not only acute phenomena linked to extreme weather conditions but also evaluate the effects of seasonality on climatic variables and air pollution. The temporal heterogeneity of the collected data, combined with the specificity of the urban context examined, makes the study area a particularly relevant case for investigating the environmental determinants of cardiorespiratory diseases.

### 2.6. Environmental Monitoring Sensors

Current technologies used for environmental monitoring are based on advanced systems that include electrochemical sensors for pollutant gases, UV photometry instruments, chemiluminescence analyzers, and optical devices for particulate matter. In particular, electrochemical sensors are used to detect carbon monoxide (CO), known for their sensitivity and selectivity, and capable of operating even at low concentrations (~ppm), making them suitable for continuous urban monitoring. Recent developments have integrated these sensors with unmanned aerial vehicle (UAV) platforms, allowing for not only spatial but also vertical monitoring of atmospheric pollutants. This approach enables us to analyze the vertical distribution of CO, PM_2.5_, PM_10_, and other gases, providing valuable data for characterizing pollution in different areas, including industrial and residential zones. Measurements show, for example, that CO concentrations are highest near the ground and tend to decrease with altitude, thus allowing for a more detailed and dynamic analysis of the pollution phenomenon beyond traditional fixed ground station monitoring [41].

For nitrogen dioxide (NO_2_), stations generally use chemiluminescence analyzers, which offer high precision and fast response times. However, recent studies also employ various types of electrochemical and optical sensors to improve coverage and measurement accuracy in different environmental contexts. For example, electrochemical sensors such as the Alphasense NO2-B43F and SPEC 3SP_NO2_5FP have been used together at several monitoring sites. These combined systems allow for more detailed and continuous detection of NO_2_ concentrations, although differences in data can depend on the sensor location and installation type, influencing sensitivity to pollution sources such as vehicular traffic or other anthropogenic activities. A comparison with reference studies has highlighted that some measurement variations are linked to instrument calibration duration and conditions, underlining the importance of accurate installation and maintenance to ensure reliable data over time [42].

Tropospheric ozone (O_3_) is detected using instruments based on UV photometry, a widely used technique for real-time monitoring due to its sensitivity and reliability even at low concentrations (ppb). However, ozone measurement still presents some challenges related to its high reactivity, its short-term temporal variability, and the difficulty in preparing stable standard solutions for calibration. Recent advances in analytical chemistry have addressed these challenges by proposing increasingly miniaturized, portable, and low-cost sensors, aiming to make O_3_ monitoring more widespread, including in indoor environments where dangerous exposures can occur from artificial sources such as air purifiers and ozone generators [43].

Microbalance sensors or equivalent technologies, which allow for the collection of highly reliable hourly and daily data, are used for PM_10_ measurements. Studies conducted in urban areas with high pollution levels, such as the city of Peshawar in Pakistan, have highlighted the importance of monitoring not only PM_10_ concentrations but also analyzing its chemical components and emission sources. PM_10_ concentrations in these areas can reach very high values, strongly influenced by meteorological factors such as temperature, relative humidity, wind speed, and precipitation. In particular, a positive correlation between PM_10_, temperature, and relative humidity has been observed, while wind and rain tend to reduce particulate concentrations. Furthermore, identifying PM_10_ sources through modeling methods has shown significant contributions from industrial emissions, soil dust and resuspension, domestic combustion, metallurgical industries, and vehicular traffic. These results emphasize the need to install integrated environmental monitoring networks that combine particulate matter and meteorological parameter measurements and to develop effective strategies for air pollution control [44].

Meteorological measurements are made using integrated digital instruments, also subject to regular quality control and maintenance, including resistance thermometers for temperature, digital barometers for atmospheric pressure, and capacitive hygrometers for estimating relative humidity. Recent technological advances have seen the adoption of complex wireless sensor models (e.g., the CWSM model) integrating different types of sensors—for temperature, humidity, pressure, solar radiation, and irradiation—enabling real-time and more detailed monitoring of environmental conditions. Such wireless systems facilitate dynamic analysis of meteorological effects in complex contexts, such as infrastructure environments, improving measurement reliability and timeliness. The implementation of advanced wireless sensor networks thus represents an important step forward to ensure precise and continuous meteorological data, essential to effectively correlate climatic conditions with air pollution phenomena [45].

All collected data were aggregated on a daily basis by calculating the arithmetic mean of hourly measurements to ensure the temporal uniformity necessary for statistical and modeling analysis. The integration of different sensor types allows for not only an accurate assessment of individual pollutants but also the analysis of possible interactions between atmospheric parameters and local meteorological conditions, a fundamental aspect for understanding the combined impact on the onset of cardiorespiratory diseases.

### 2.7. ARPA Puglia Monitoring Network

The monitoring of the environmental and climatic variables underlying this study was carried out by ARPA Puglia through an integrated network of fixed monitoring stations strategically distributed across the Bari area and the wider region. This network includes the Regional Air Quality Network (RRQA) and an automatic meteorological network, both composed of stations equipped with certified sensors that are regularly calibrated and designed to ensure accuracy, precision, and repeatability in measurements. The technologies used enable real-time monitoring of a wide range of atmospheric parameters, including gaseous pollutants (carbon monoxide (CO), nitrogen dioxide (NO_2_), and tropospheric ozone (O_3_)), particulate matter (PM_10_), and meteorological indicators such as air temperature, atmospheric pressure, relative humidity, wind speed, and wind direction. The five main stations (located in Bari, Brindisi, Foggia, Lecce, and Taranto) also measure solar radiation and, at some sites, the UV index using UV-E radiometers and LPUVI02AV instruments. In addition to these, twenty supplementary stations connected to the air quality network provide meteorological data at an hourly resolution. The meteorological sensors used by ARPA Puglia are state-of-the-art and include ultrasonic anemometers, global radiation sensors, and high-resolution digital rain gauges. All data are validated according to the guidelines of the National System for Environmental Protection (SNPA) and are made publicly available through WebGIS platforms, ensuring transparency and scientific quality in the data collection process [46].

### 2.8. Data Sources, Code Development, and Ethical Considerations

In this section, we describe the origin and nature of the data used, the software implementation of the machine learning models, and the ethical aspects considered in this study.Clinical data: Provided by the Policlinico of Bari, consisting of anonymized records of emergency department visits.Environmental data: Obtained from ARPA Puglia [46], which operates regional meteorological and air quality monitoring stations.Code development: The models were developed and tested using the Python 3.11.11 programming language. The main libraries used include:○pandas and numpy for data handling and manipulation;○scikit-learn (modules: model_selection, metrics, preprocessing, ensemble) for:-data splitting (e.g., train_test_split);-performance evaluation through cross-validation (cross_val_score);-computation of metrics (e.g., MAE, RMSE);-preprocessing (e.g., normalization via MinMaxScaler or StandardScaler);-model training with RandomForestRegressor;-xgboost (XGBRegressor) and lightgbm (LGBMRegressor) for advanced boosting techniques;-interpret (glassbox module) for the Explainable Boosting Machine (EBM);-Optuna for automated Bayesian hyperparameter optimization;-joblib for saving and loading models and experimental outputs;-shap and lime for global and local model interpretability;-matplotlib and seaborn for visualization and exploratory analysis.

Ethical considerations:○Only fully anonymized and aggregated data were used.○No personal identifiers are present, and no patient can be traced.○No human or animal experiments were performed.○As such, ethical approval was not required according to applicable regulations.
Generative Artificial Intelligence (GenAI) tools were used solely to assist with partial language editing.

## 3. Results

### 3.1. Application to the Case Study

#### 3.1.1. Meteorological Data

Regarding the climatic variables used in this study, the following were considered: average air temperature (Tavg) (°C), dew point temperature (DEW) (°C), atmospheric pressure (P) (hPa), and relative humidity (RH) (%). As for the environmental parameters, carbon monoxide (CO) (mg/m^3^), fine particulate matter (PM_10_) (µg/m^3^), nitrogen dioxide (NO_2_) (µg/m^3^), and ozone (O_3_) (µg/m^3^) were included. The period covered extends over eleven years, from 2013 to 2023, providing a solid basis for the analysis of long-term dynamics. To describe the characteristics of the collected data, various statistical parameters were calculated, including the minimum, maximum, mean, standard deviation, median (corresponding to the 50th percentile), first and third quartiles (25th and 75th percentiles, respectively), and 90th, 95th, and 99th percentiles. These measures allowed for a detailed overview of the temporal distribution of the variables, highlighting potential outliers (Table 2).

In addition, to enrich the temporal component of the analysis, a script was developed to extract additional information from the dates included in the dataset. Specifically, seasonal components (e.g., summer, winter) were calculated, the day of the week for each observation (e.g., Monday, Tuesday) was identified, and a binary variable was added to indicate the presence of public holidays according to the Italian national calendar. These factors were included as explanatory features in the model, as they can significantly influence environmental exposure levels, individual behaviors, and, consequently, the trend of hospital admissions for cardiorespiratory diseases.

#### 3.1.2. Emergency Room (ER) Data

The health data used in this study refer to patients affected by cardiorespiratory diseases (CRDs) and were obtained from the Policlinico of Bari, one of the main hospitals in southern Italy. The considered diagnoses include dyspnea, chest pain, precordial pain, cardiac rhythm disorders, arterial hypertension, and other general respiratory conditions. The integration of this clinical information with meteorological and environmental variables allows for an in-depth investigation of potential correlations between air pollution, climatic conditions, and the cardiorespiratory health of the population in the Bari metropolitan area. This synergy between heterogeneous data sources is a key element in identifying recurring patterns and preventing acute health events related to critical environmental conditions.

#### 3.1.3. Application of Machine Learning Algorithms

First of all, the optimization of model parameters was performed. Table 3 shows the optimal parameters for each model identified by Bayesian optimization with Optuna and three-fold cross-validation on the training set.

At this stage, the focus was placed on evaluating the performance of the four models, which was quantified using standard regression metrics (the coefficient of determination (R^2^), mean absolute error (MAE), mean squared error (MSE), and root mean squared error (RMSE)). The average results obtained enabled an objective comparison of the behavior of the four algorithms. Table 4 reports the performance of the three models during the test phase.

The training of the models showed that all four algorithms considered presented solid performance. The Random Forest model achieved a coefficient of determination (R^2^) of 0.881 during the test phase, indicating good adaptability and accuracy in simulating cases of CRDs on previously unseen data. The mean absolute error (MAE) was 0.051 cases per day, demonstrating a low simulation error.

The XGBoost model exhibited even more competitive performance, with an R^2^ of 0.901 and a lower MAE (0.047 cases/day), highlighting greater accuracy compared with Random Forest. LightGBM also showed strong performance, with an R^2^ of 0.896 and an MAE of 0.048 cases/day, confirming its good predictive reliability.

In addition to the MAE and R^2^, the mean squared error (MSE) and its square root (RMSE) were also analyzed, as they are useful metrics for evaluating the dispersion of errors. The MSE and RMSE values remained low across all models, confirming a fair degree of consistency in the distribution of simulation errors.

Based on this first analysis, the XGBoost model showed the best trade-off between accuracy and predictive stability, with the lowest absolute error values among those obtained.

In order to objectively compare the predictive performance of the developed machine learning models, a statistical significance analysis was conducted on the distributions of the scores obtained during cross-validation through the application of the t-Student test. Since pairwise comparisons using this test require the data distributions to be approximately normal, this assumption was preliminarily verified using the Kolmogorov–Smirnov test [47,48]. All models considered (XGBoost, Random Forest, LightGBM, and Explainable Boosting Machine (EBM)) showed distributions compatible with normality (*p*-value > 0.05), thus making the use of the *t*-test appropriate for evaluating the significance of the observed differences. The t-Student test was applied to the distributions of the coefficient of determination (R^2^) obtained on the test set in order to evaluate the significance of the differences in predictive performance between the models. The results of this analysis are reported in Table 5.

The *t*-test results indicate that the differences in predictive performance between XGBoost and each of the other models (Random Forest, LightGBM, and EBM) are statistically significant (*p*-value < 0.001). This finding confirms that XGBoost consistently outperforms the competing models in forecasting daily emergency room visits for cardiorespiratory conditions. Therefore, the statistical analysis provides strong evidence supporting the superiority of XGBoost for this specific predictive task.

Following the analysis of the predictive performance of the models, conducted using both quantitative metrics and statistical tests to assess their significance, a broader evaluation was carried out by comparing the results of this study with those reported in the existing literature. This comparative analysis, presented in Table 5 of the Discussion section, reinforces the validity and robustness of the approach adopted within the context of similar epidemiological predictive tasks.

The analysis was further enriched by visualizing the errors in a comparison plot between actual values and those predicted by the algorithm (Figure 2).

On the plot, error bands of ±10% and ±20% (in green and light blue, respectively) were drawn. The analysis showed that the vast majority of points (92%) fall within the ±20% bands, indicating a low incidence of significant errors (8.2%). Moreover, a high density of points (83%) is concentrated around the bisector and falls within the narrower ±10% band, suggesting a high level of accuracy in the meta-model’s predictions.

#### 3.1.4. Global and Local Interpretability (SHAP and LIME Methods)

Since XGBoost demonstrated the best performance, its interpretability was further explored using the SHAP (SHapley Additive exPlanations) method. Specifically, the global importance of the features was analyzed using a bar chart of the mean absolute SHAP values (Figure 3). This representation allows for a concise quantification of the average contribution of each variable to the model’s predictions.

Since the most significant variables are those whose cumulative contribution to the model’s prediction reaches approximately 70–80% [49], the analysis of the histogram in Figure 3 reveals the top four most influential features: carbon monoxide, contributing 31%, atmospheric pressure (26%), average temperature (13%), and relative humidity (7%), for a total cumulative contribution of 77%.

To gain a deeper understanding of the relationships between features and the model output, a global Bee Swarm plot was then generated, as shown in Figure 4.

This plot enables a global interpretation of the results obtained by showing, for each variable, both the range of values (encoded using a color scale from blue to red) and the effect of those values on the prediction (in terms of SHAP value). From the Bee Swarm plot, the following insights emerged:CO has a positive effect on the target: High concentrations of CO (in red) are associated with positive SHAP values, suggesting an increased likelihood of CRD cases.P and Tavg show an inverse impact: Low values (in blue) are associated with positive SHAP values, implying that low pressure and low temperature conditions contribute to an increase in cases.RH displays a pattern similar to that of carbon monoxide, with higher values linked to a positive impact on the target.

To further validate and analyze these findings in greater detail, individual Bee Swarm plots were generated for each of the four selected variables (Figure 5). In these plots, the X-axis represents the actual values of the variable, while the Y-axis shows the corresponding SHAP values.

As can be observed:For CO (Figure 5a), high values (approximately greater than 0.75–0.80 mg/m^3^) are associated with positive SHAP values and a strong red coloration, confirming the role of this variable as a risk factor when concentrations are elevated.For P_atm (Figure 5b) and Tavg (Figure 5c), the relationship is inverse, as already noted from the global Bee Swarm plot. Lower values tend to be associated with higher SHAP values, highlighting how harsher atmospheric conditions are more favorable to an increase in CRD cases (critical thresholds appear to be around 1010 hPa for P_atm and below approximately 20 °C for Tavg).For RH (Figure 5d), the same trend observed for CO emerges again, with high humidity levels (around 70% or more) showing a positive effect on the target.

After exploring the global influence of the variables using the SHAP method, a local analysis was conducted using LIME (Local Interpretable Model-agnostic Explanations). This approach is particularly useful for understanding in detail and at the individual observation level how each of the four variables influences the prediction [32].

For each of them, the local influence was analyzed to identify threshold values above or below which there is a relevant impact on the number of daily CRD cases. The results of this analysis are summarized in four explanatory histograms (Figure 6a–d), which display the identified value ranges.

In the charts shown in Figure 6, the X-axis represents the “Mean LIME Impact on CRDs”, which quantifies how much and in what direction (positive or negative) each specific value range affects the model’s predictions. The Y-axis displays the different value intervals for each of the four analyzed variables.

By closely examining the charts and focusing only on the positive portion of the X-axis (corresponding to an increase in CRD cases), the following observations emerge:Significant increases in CRD cases were observed when CO levels exceeded 0.84 mg/m^3^.CRD cases increased when P_atm values were ≤1006.81 hPa.An increase in CRD numbers is mainly associated with average temperatures (Tavg) between 12.28 °C and 17.19 °C, which represent the interval with the highest incidence. However, even temperatures ≤ 12.28 °C show a positive association, though with a less significant contribution.Relative humidity (RH) values between 70.33% and 75.79% contribute more substantially to the rise in CRD cases, whereas levels > 75.79%, although still linked to a positive effect, have a lesser impact.

## 4. Discussion

The threshold values qualitatively deduced using the SHAP method were confirmed in the LIME analysis, which enabled a numerical quantification. Thanks to the local decomposition of the predictions, LIME made it possible to identify the intervals of each variable associated with a significant impact on the increase in CRD cases. Therefore, the condition that maximizes the increase in CRD cases occurs when the following critical intervals are simultaneously present:CO > 0.84 mg/m^3^;P_atm ≤ 1006.81 hPa;Tavg ≤ 17.19 °C;RH > 70.33%.

Following the identification of these intervals, experimental verification of the obtained results was carried out. It was verified whether the condition defined by the simultaneous occurrence of the four calculated intervals could identify all CRD values exceeding a certain threshold. This threshold was set at the 95th percentile. In Figure 7, these elements are represented, showing that all values above the defined threshold occur on dates corresponding to the simultaneous occurrence of the four intervals of meteorological and pollution variables considered most important by the SHAP feature importance analysis.

These results strengthen the interpretative robustness of machine learning models, especially when combined with XAI models such as SHAP and LIME, confirming the relevance of the identified intervals as critical environmental thresholds. Therefore, the analysis provides useful support for predicting peaks in hospital admissions and can serve as an operational basis for decision-making tools used by health and environmental authorities, aimed at the timely activation of alert or containment measures in the presence of unfavorable atmospheric conditions.

The results of the present study are consistent with emerging evidence on the complex interaction between meteorological factors and air pollution in influencing the incidence of cardiorespiratory diseases. Recent studies highlight how environmental variables such as temperature and relative humidity, in combination with specific pollutants, increase the risk of hospital admissions for cardiovascular conditions [6]. Moreover, a broader review underscores the growing role of pollution and climate change as key contributors to the spread of non-communicable chronic diseases [14]. This work contributes to this framework by identifying operational environmental thresholds that are useful for the prevention and management of health impacts.

The results obtained in this study can be compared in terms of performance with those reported in the literature (Table 6).

Compared with the existing literature, the present study demonstrates superior predictive performance. For instance, Lu et al. (2021) [10] applied ARIMA, MLP, and LSTM models to forecast hospital visits for respiratory diseases. The highest R^2^ reached 0.80 for the MLP model, while LSTM achieved 0.78. Regarding the mean absolute error (MAE), ARIMA exhibited the largest average error, with approximately 99 mispredicted cases per day. The MLP model reduced this error to 49 cases per day, and the LSTM model performed best with an MAE of 33 cases per day. These MAE values represent the average number of hospital visits incorrectly predicted daily [10].

Ravindra et al. (2023) [11] evaluated multiple machine learning models on outpatient visits for acute respiratory infections. The Random Forest (RF) model reached an R^2^ of 0.606 for ARI patients and up to 0.872 across the entire dataset. However, exact MAE values were not reported, leaving the magnitude of prediction errors unspecified [11].

Kurucz et al. (2024) [12] utilized multiple linear regression (MLR) and Random Forest models to predict emergency visits for acute coronary syndrome. These models achieved R^2^ values up to 0.80 for unstable angina cases. In terms of MAE, errors were reported as up to 7.8 cases per day on average overall and 5.3 cases per day specifically for unstable angina, indicating the average number of mispredicted emergency visits per day. The table does not clarify which model these MAE values refer to, but they provide a general indication of the prediction error magnitude for the dataset [12].

Hu et al. (2020) [13] developed a stacking ensemble combining LR, SVR, XGBoost, RF, and GBDT to predict daily hospital admissions for cardiovascular diseases. The stacking model achieved an R^2^ equal to 0.90 and an MAE equal to 20.69, demonstrating a consistent and modest enhancement in prediction accuracy [13].

In comparison, the XGBoost model selected in this study achieved an R^2^ of 0.901 and an exceptionally low MAE of 0.047 cases/day, significantly outperforming previously reported models in both accuracy and error metrics. This suggests that the present approach not only effectively captures the complex interactions between environmental and meteorological variables but also provides a robust and reliable predictive tool for local healthcare management.

Although the model architecture is replicable, its application to other geographical contexts would require careful recalibration based on local environmental and healthcare data. Predictive performance could be influenced by variations in climatic determinants or differences in healthcare system responses. Moreover, in addition to the 10-fold cross-validation performed to ensure the internal robustness of the model, external validation—using data from different cities or time periods—would be essential to rigorously assess the model’s generalizability beyond the metropolitan area of Bari. Due to current data availability constraints, such validation was not conducted in this study but is planned for future work.

In continuing to explore future developments, the integration of additional environmental variables or the combination of meteorological–hydrological models with machine learning approaches is hypothesized in order to improve early warning systems for cardiorespiratory risk. A more detailed discussion of future perspectives is provided in the Section 5.

## 5. Conclusions

This work aimed to develop an interpretable machine learning model to estimate the daily incidence of cardiorespiratory diseases based on meteorological and pollution variables. To this end, a benchmarking approach was adopted, comparing the performance of four regression models: Random Forest, Extreme Gradient Boosting, Light Gradient Boosting Machine, and Explainable Boosting Machine (EBM). The dataset, spanning eleven years (2013–2023), integrated three main information sources:Meteorological data (mean temperature, atmospheric pressure, relative humidity, etc.);Environmental data (pollutants such as CO, NO_2_, and PM10);Health data on daily emergency room admissions for cardiorespiratory conditions in the Bari area.

Environmental and meteorological measurements were obtained through fixed stations equipped with certified digital sensors, based on technologies such as UV photometry for ozone, microbalances for particulates, and electrochemical sensors for pollutant gases. These devices, regularly calibrated and quality-checked, ensure continuous, reliable, and high-resolution temporal data collection. The availability of accurate data for both atmospheric and meteorological variables is essential for building robust predictive models. The dataset was also enriched with date-derived information, such as seasonal components (e.g., winter, summer), to help explain variations in cardiorespiratory hospitalizations.

Among the four models evaluated on the test set, XGBoost proved to be the most reliable, with an R^2^ of 0.901 and an MAE of 0.047 cases/day, demonstrating strong generalization and adaptability to the problem. Consequently, subsequent global and local interpretability analyses were conducted on the XGBoost model.

The use of SHAP allowed for the identification of the most relevant environmental and climatic variables in the model: carbon monoxide (CO), atmospheric pressure (P_atm), mean temperature (Tavg), and relative humidity (RH), as supported by the global Bee Swarm plot. Individual Bee Swarm plots provided preliminary qualitative estimates of critical ranges associated with increased CRD admissions. Applying LIME, followed by a careful verification phase, precisely identified the environmental thresholds associated with significant increases in cardiorespiratory disease cases. Specifically, the CRD incidence significantly increased when the following conditions occurred simultaneously:CO concentrations exceed 0.84 mg/m^3^;P_atm is less than or equal to 1006.81 hPa;Tavg is less than or equal to 17.19 °C;RH exceeds 70.33%.

These ranges represent operational thresholds that define potentially critical environmental conditions, in correspondence with which a marked increase in hospital admissions for cardiorespiratory causes was observed.

From a methodological standpoint, an important study phase involved verifying the results. This procedure verified the model’s ability to identify daily CRD case peaks that exceed the 95th percentile of the CRD distribution during 2013–2023. The results show that the model effectively captures these critical conditions, demonstrating a high level of reliability in identifying the most significant health events.

In conclusion, the integrated approach of global and local variable analysis, combined with the ability to translate results into applicable and observable thresholds, confirms the model’s effectiveness not only as an epidemiological tool but also as a form of operational support for prevention. In the future, using such interpretable models could offer valuable contributions to health and environmental surveillance systems, providing timely and objective indications for activating alert or containment measures in the presence of adverse atmospheric conditions.

Among future development prospects, one notable possibility is to integrate meteorological–hydrological models with machine learning models. Such synergy would allow for earlier identification of critical health alert thresholds, improving public health impact management and strengthening prevention strategies. This evolution would also open new research and practical application opportunities in the medical and environmental sectors.

Moreover, extending the study to multiple centers or geographical areas would represent an important future development, useful for adapting the model to different epidemiological and environmental scenarios. Complementing this, an external validation phase is planned on independent datasets from urban contexts and different time periods in order to rigorously assess the stability and generalizability of the model.

## 6. Patents

No patents have been filed to date. However, potential applications of the developed methods, particularly in the area of early warning systems, may be considered for future patenting.

## Figures and Tables

**Figure 1 sensors-25-04864-f001:**
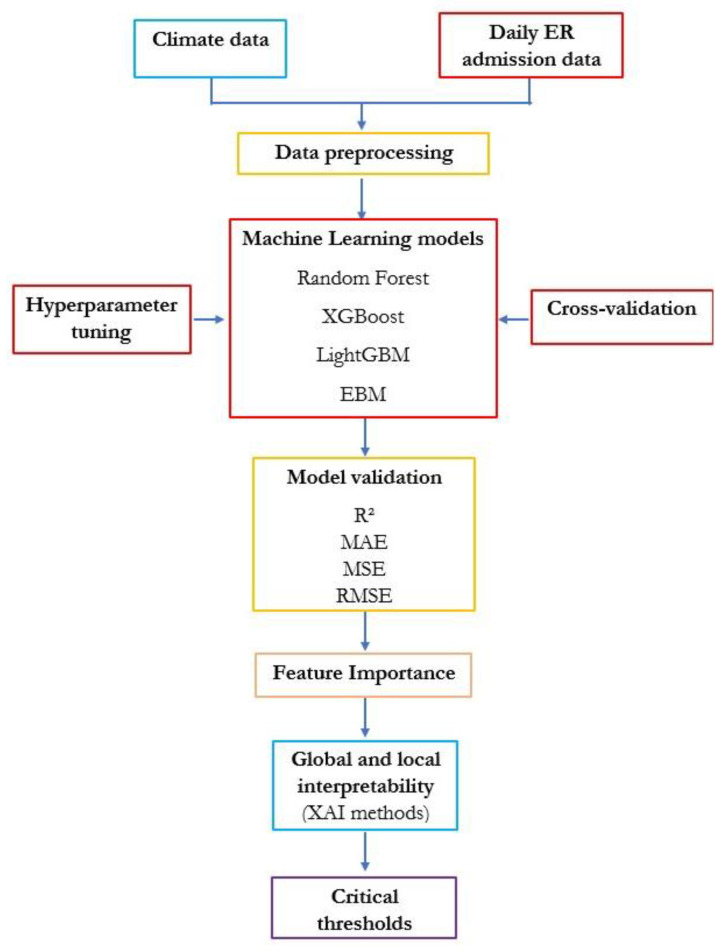
Methodological flow chart.

**Figure 2 sensors-25-04864-f002:**
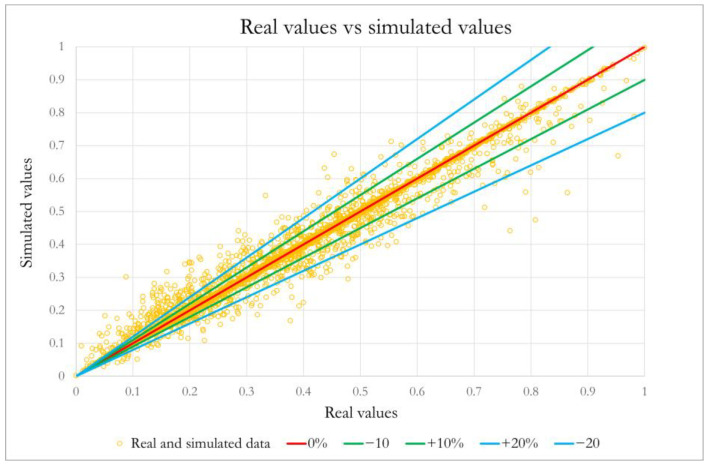
Scatter plot of observed vs. predicted values by the XGBoost model.

**Figure 3 sensors-25-04864-f003:**
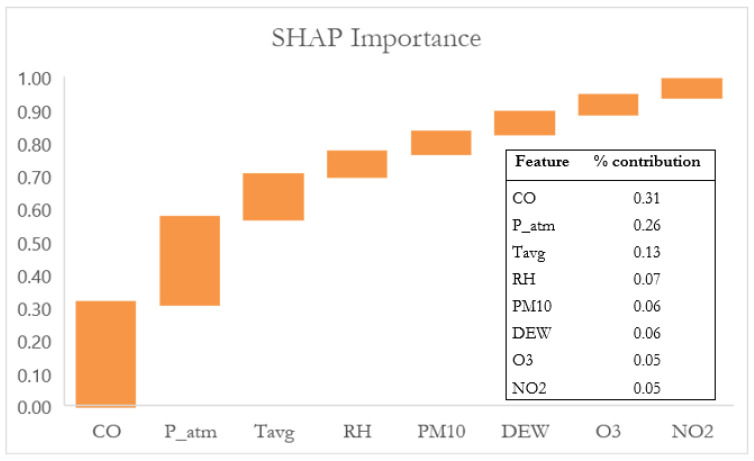
Global SHAP feature importance histogram.

**Figure 4 sensors-25-04864-f004:**
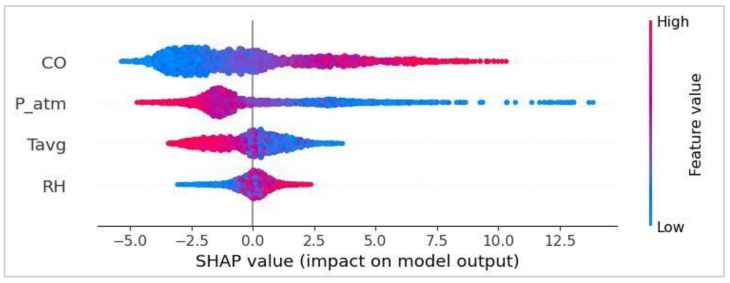
Global Bee Swarm plot of the top four features.

**Figure 5 sensors-25-04864-f005:**
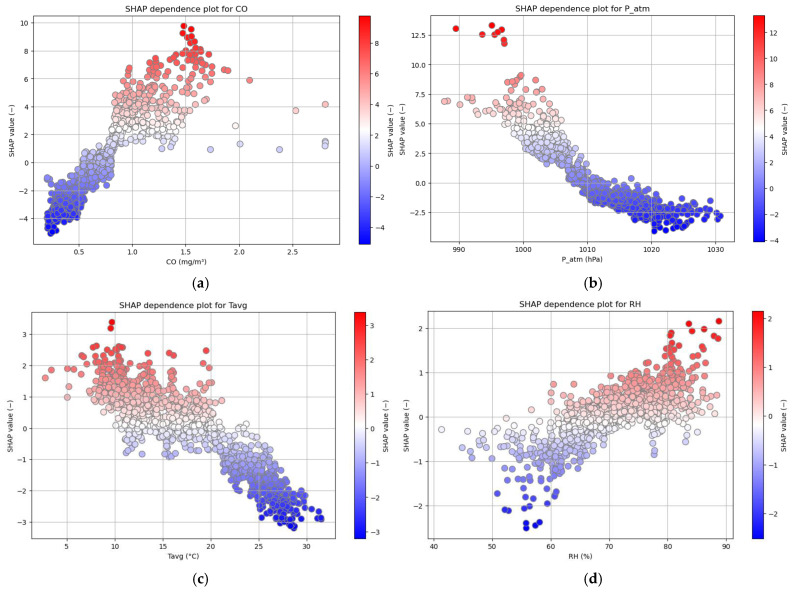
Individual SHAP Bee Swarm plots for the top four features. (**a**) CO; (**b**) P_atm; (**c**) Tavg; (**d**) RH.

**Figure 6 sensors-25-04864-f006:**
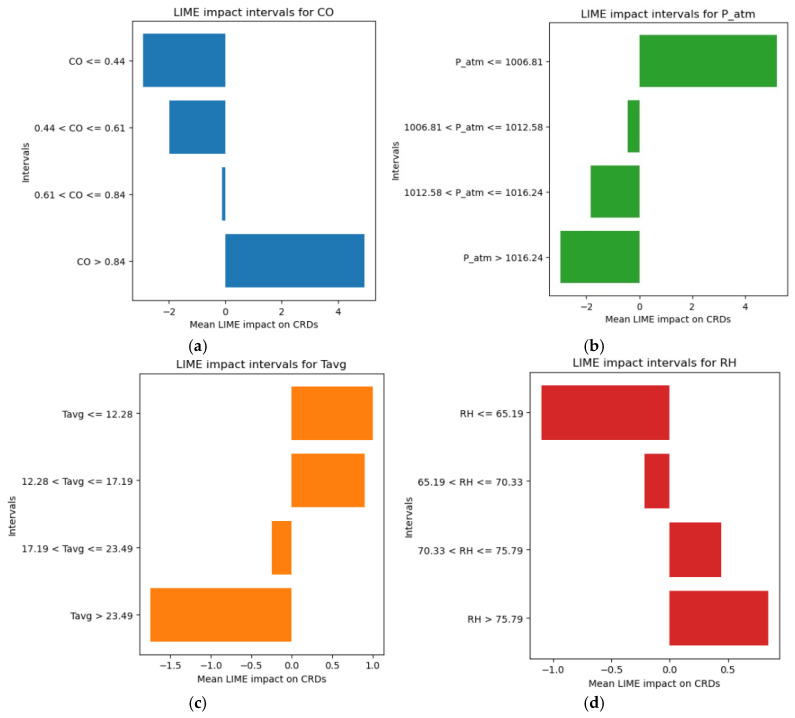
LIME-based interpretability of CRD risk by feature value ranges. (**a**) CO; (**b**) P_atm; (**c**) Tavg; (**d**) RH.

**Figure 7 sensors-25-04864-f007:**
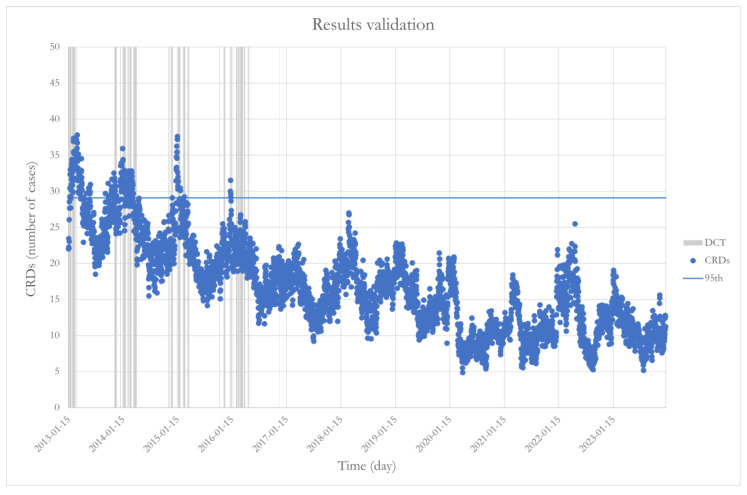
Temporal trend of CRD hospital admissions with percentile values and the dates corresponding to the critical thresholds (DCTs)**.**

**Table 1 sensors-25-04864-t001:** Machine learning models and findings from selected environmental health studies.

Title	Authors	Machine Learning Models Used	Objective	Key Findings
**Application of machine learning approaches to predict the impact of ambient air pollution on outpatient visits for acute respiratory infections**	Ravindra, Katoch, Mor et al. (2023) [11]	RF, KNN, LR, LASSO, DT, SVR, XGBoost, DNN	Predict outpatient visits for respiratory infections using air pollution data	Air pollutants showed significant predictive power; Random Forest yielded the best performance
**Prediction of emergency department presentations for acute coronary syndrome using a machine learning approach**	Kurucz, Schenk, Veelo et al. (2024) [12]	MLR, RF	Forecast ED visits for acute coronary syndrome using meteorological variables	Seasonal and daily patterns identified; strong predictive accuracy for UA and NSTEMI (R^2^ = 0.80)
**A Stacking Ensemble Model to Predict Daily Number of Hospital Admissions for Cardiovascular Diseases**	Hu, Qiu, Su et al. (2020) [13]	LR, SVR, XGBoost, RF, GBDT, Stacking	Predict daily hospitalizations for CVD using an ensemble of base models	The stacking model outperformed all base models in terms of MAE, RMSE, MAPE, and R^2^
**Interactive short-term effects of meteorological factors and air pollution on hospital admissions for cardiovascular diseases**	He, Zhai, Liu et al. (2022) [6]	Generalized Additive Models (GAMs) with interactions	Examine the interactive short-term effects of meteorological factors and pollutants on CVD	Synergistic effects found between cold temperatures and PM pollution, particularly among elderly patients
**Feasibility of machine learning methods for predicting hospital emergency room visits for respiratory diseases**	Lu, Bu, Xia, Lu, Yao, Jiang (2021) [10]	ARIMA, MLP, LSTM	Forecast ER visits for respiratory conditions based on PM_2.5_ exposure in Beijing	LSTM and MLP showed strong predictive performance; PM_2.5_ was a critical risk factor
**Environmental risk factors and cardiovascular diseases: a comprehensive expert review**	Münzel, Hahad, Lelieveld et al. (2021) [14]	Narrative Review	Analyze physical and psychosocial environmental factors influencing cardiovascular disease	Identified pathophysiological mechanisms; proposed mitigation strategies for air, noise, and light pollution

**Table 2 sensors-25-04864-t002:** Statistical description of the data (EMA7).

	Tavg	DEW	P_atm	RH	CO	O3	PM10	NO2	CRDs
**max**	31.7	23.7	1031.3	90.7	2.8	134.9	74.8	112.8	37.8
**min**	2.8	−3.6	987.8	39.8	0.1	40.8	7.4	19.0	4.9
**avg**	17.8	12.1	1011.8	70.3	0.7	82.8	22.7	53.0	16.8
**std**	6.2	5.3	6.9	7.6	0.4	18.4	7.2	14.8	6.4
**99th**	29.6	21.4	1027.2	86.1	1.7	122.4	44.0	93.8	33.6
**95th**	27.6	20.2	1022.9	82.1	1.4	111.4	36.1	79.3	29.1
**90th**	26.4	19.1	1020.1	80.1	1.2	106.3	32.2	72.9	25.9
**75th**	23.5	16.7	1016.2	75.8	0.8	97.3	26.5	62.0	20.9
**50th**	17.2	12.1	1012.5	70.4	0.6	82.9	21.4	51.4	16.4
**25th**	12.3	7.7	1006.6	65.2	0.4	67.9	17.6	43.0	11.5

**Table 3 sensors-25-04864-t003:** Optimized parameters of models using Bayesian optimization and three-fold CV on the training set.

Optuna Optimization					
Hyperparameter	Range	Adopted Value	Hyperparameter	Range	Adopted Value
*XGBoost*			*Random Forest*		
learning_rate	0.01–0.3	0.05	n_estimator	100–1000	995
max_depth	3–10	8	max_depth	3–30	20
n_estimator	100–1000	645	min_sample_split	2–10	4
subsample	0.2–1.0	0.43			
*LightGBM*			*EBM*		
n_estimator	100–1000	717	max_bins	64–512	429
max_depth	−1–30	−1	max_interact_bins	32–256	208
learning_rate	0.01–0.3	0.07	interactions	0–10	10
num_leaves	20–300	128	learning_rate	0.01–0.3	0.1
subsample	0.5–1.0	0.91	min_samples_leaf	2–50	46
colsample_bytree	0.5–1.0	0.9	max_leaves	2–64	8

**Table 4 sensors-25-04864-t004:** Performance of the four models on the test set.

Random Forest	XGBoost	LightGBM	EBM
R^2^ (−) = 0.881	R^2^ (−) = 0.901	R^2^ (−) = 0.896	R^2^ (−) = 0.813
MAE (case/day) = 0.051	MAE (case/day) = 0.047	MAE (case/day) = 0.048	MAE (case/day) = 0.066
MSE (−) = 0.005	MSE (−) = 0.004	MSE (−) = 0.004	MSE (−) = 0.007
RMSE (−) = 0.068	RMSE (−) = 0.062	RMSE (−) = 0.063	RMSE (−) = 0.085

**Table 5 sensors-25-04864-t005:** Comparison of model performance using the t-Student test.

Model 1	Model 2	*p*-Value	Significant
XGBoost	Random Forest	<0.001	Yes
XGBoost	LightGBM	<0.001	Yes
XGBoost	EBM	<0.001	Yes

**Table 6 sensors-25-04864-t006:** Comparison of model performance between related studies and the current study.

Article Title	ML Models Used	Target	R^2^ (Test)	MAE (Test) (Cases/Day)
**Feasibility of machine learning methods for predicting hospital emergency room visits for respiratory diseases** *(Lu, Bu, Xia* et al.*, 2021)* [10]	ARIMA, MLP, LSTM	Hospital visits for respiratory diseases	ARIMA: 0.70 MLP: 0.80 LSTM: 0.78	ARIMA: 99 MLP: 49 LSTM: 33
**Application of machine learning approaches to predict the impact of ambient air pollution on outpatient visits for acute respiratory infections** *(Ravindra, Katoch, Mor* et al.*, 2023)* [11]	RF, KNN, LR, LASSO, DT, SVR, XGBoost, DNN	Outpatient visits for acute respiratory infections (ARIs)	RF: 0.61 (ARI) 0.87 (total patients)	n.a.
**Prediction of emergency department presentations for acute coronary syndrome using a machine learning approach** *(Kurucz, Schenk, Veelo* et al.*, 2024)* [12]	MLR, RF	Emergency visits for acute coronary syndrome (ACS)	MLR: 0.66 (overall) 0.80 (unstable angina)	7.8 (overall) 5.3 (unstable angina)
**A Stacking Ensemble Model to Predict Daily Number of Hospital Admissions for Cardiovascular Diseases** *(Hu, Qiu, Su* et al.*, 2020)* [13]	LR, SVR, XGBoost, RF, GBDT, Stacking	Hospital admissions for cardiovascular diseases	Stacking: 0.90	Stacking: 20.69
**[Current Study] Machine Learning interpretability to analyze the interaction between cardiorespiratory diseases and** **meteo-pollutant sensor data**	XGBoost (selected among RF, LightGBM, and EBM)	Hospital admissions for cardiorespiratory diseases	XGBoost: 0.90	XGBoost: 0.05

## Data Availability

The data supporting the findings of this study were provided by ARPA Puglia and Policlinico of Bari. Due to privacy and ethical restrictions, the original datasets are not publicly available. However, statistical summaries are reported within the article. Researchers interested in accessing the data may contact the corresponding author for reasonable requests, subject to approval by the data providers.

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
