# Peer review of "An Interpretable Machine Learning Framework for Analyzing the Interaction Between Cardiorespiratory Diseases and Meteo-Pollutant Sensor Data"

_sensors, 2025, doi:10.3390/s25154864_

Round 1
Reviewer 1 Report
Comments and Suggestions for Authors
A well-written paper investigates the integration of artificial intelligence in healthcare. Considering the challenges and impact of AI in healthcare, it is a good research that can provide better support to the healthcare industry. Here are some suggestions for the authors to improve it and make it more understandable for the reader:
- Introduction:
- The introduction is good, but this section failed to sufficiently articulate why existing work is inadequate and what specific gaps this research aimed to fill. Authors need to discuss more about specific gaps.
- In addition, research motivation and major findings should be clearly mentioned in the introduction section.
- It can also be good to add a paragraph at the end discussing the structure of the paper.
- Related Work: The paper currently lacks a literature review or related work section, which is quite unusual. The authors should include a dedicated section discussing the state-of-the-art research in this domain. Additionally, a table critically analysing previous methodologies would help highlight the research gap and further justify the significance of the proposed work.
- Methodology:
- Authors should explain each evaluation metric with its impact on analysing the results.
- 5 Data sources, code development and ethical considerations: Why are there two-line paragraphs? It would be good to use bullet points here or increase the size of the paragraph.
- Results and Discussion:
- The discussion is weak. It should have more details about the major findings. Specifically, comparing the work with existing work needs to be done. A table can be added to compare the work with existing models’ performance.
Author Response
Comment 1. A well-written paper investigates the integration of artificial intelligence in healthcare. Considering the challenges and impact of AI in healthcare, it is good research that can provide better support to the healthcare industry. Here are some suggestions for the authors to improve it and make it more understandable for the reader:
Introduction: The introduction is good, but this section failed to sufficiently articulate why existing work is inadequate and what specific gaps this research aimed to fill. Authors need to discuss more about specific gaps. In addition, research motivation and major findings should be clearly mentioned in the introduction section. It can also be good to add a paragraph at the end discussing the structure of the paper.
Response 1. We sincerely thank you for the valuable suggestions. The introduction has been expanded to include a more detailed discussion of the specific gaps in the existing literature and the motivation behind the present study, highlighting the importance of machine learning models that balance predictive accuracy and interpretability. Additionally, a final paragraph has been added to outline the structure of the paper, to facilitate its readability.
Comment 2. Related Work: The paper currently lacks a literature review or related work section, which is quite unusual. The authors should include a dedicated section discussing the state-of-the-art research in this domain. Additionally, a table critically analysing previous methodologies would help highlight the research gap and further justify the significance of the proposed work.
Response 2. We thank the Reviewer for pointing out the lack of a dedicated section on related work. In response, a new section titled “State of the Art” has been added, providing a detailed review of the most relevant studies that have applied machine learning techniques to analyze the environmental impact on hospital admissions for cardiovascular and respiratory diseases. This section also includes a summary table of the reviewed literature, outlining the models used, the authors’ objectives, and the main findings, with the aim of identifying existing gaps and justifying the relevance of the present work.
Comment 3. Methodology: Authors should explain each evaluation metric with its impact on analyzing the results. 5 Data sources, code development and ethical considerations: Why are there two-line paragraphs? It would be good to use bullet points here or increase the size of the paragraph.
Response 3. We thank the Reviewer for the helpful comment. In response, the following improvements have been made:
•In Section 2.3, the description of the evaluation metrics (R², MAE, MSE, RMSE) has been expanded to more clearly explain their meaning and their impact on the interpretation of the results.
• In Section 2.8, the content regarding data sources, code development, and ethical aspects has been reformulated in a more comprehensive and structured manner, also using bullet points to improve readability and completeness.
Comment 4. Results and Discussion: The discussion is weak. It should have more details about the major findings. Specifically, comparing the work with existing work needs to be done. A table can be added to compare the work with existing models’ performance.
Response 4. We thank the Reviewer for the valuable suggestion, which has helped enhance the overall quality of the “Discussion” section. In response, this section has been thoroughly revised and expanded, including a more in-depth analysis of the results in terms of predictive performance. Moreover, the obtained results have been contextualized in light of the existing scientific literature and compared—also using an appropriate table—with the performance reported in recent studies.

Reviewer 2 Report
Comments and Suggestions for Authors
1. Abstract: Briefly contextualize the problem to be solved.
2. Why were Random Forest, XGBoost, LightGBM, and Explainable Boosting Machine used? It would be interesting to compare them with other methods such as Support Vector Machines, Artificial Neural Networks, etc.
3. Explain the use of generative AI in more detail.
4. Which Python libraries were used?
5. Which parameters of the machine learning methods were optimized? What was the search interval?
6. What were the parameters of the Bayesian optimization method?
7. What are the optimal parameters?
8. How many iterations with different random seeds were performed?
9. Was a K-Fold or something similar applied along with Bayesian optimization?
10- Can be applied a statistical test to verify the differences are significant statistically.
Author Response
Comment 1. Abstract: Briefly contextualize the problem to be solved.
Response 1. We thank you for the constructive comment, which allows us to clarify this aspect of the study more effectively.
This study focuses on analyzing the impact of meteorological and environmental factors on the onset of cardiorespiratory conditions that lead patients to seek emergency care. Using regression models based on climate, pollution, and hospital data, the aim is to predict the daily number of cardiorespiratory cases in order to optimize healthcare resource management and identify the most influential environmental parameters. As suggested, this description has been incorporated and clarified in the abstract to provide a completer and more accurate summary of the work.
Comment 2. Why were Random Forest, XGBoost, LightGBM, and Explainable Boosting Machine used? It would be interesting to compare them with other methods such as Support Vector Machines, Artificial Neural Networks, etc.
Response 2. We are grateful for the observation, which provided an opportunity to improve the manuscript.
Initially, six different models were tested—XGBoost, Random Forest, LightGBM, Explainable Boosting Machine, Ridge, and TabNet—to evaluate their predictive capabilities in analyzing the influence of environmental and meteorological factors on the onset of cardiorespiratory diseases. Preliminary results clearly showed that the Ridge and TabNet models performed significantly worse than the other four in terms of both the coefficient of determination (R²) and test set error metrics.
Support Vector Machines (SVMs) were not considered in this study because, although they can perform well on small datasets, they entail high computational costs when applied to large-scale data such as those used here. In addition, they require extensive preprocessing to handle heterogeneous numerical variables effectively and, compared to other algorithms used, offer limited interpretability.
Based on these considerations, we decided to focus the final comparison on the four best-performing models. This choice was driven by the goal of ensuring a more robust and meaningful analysis, emphasizing models capable of providing scientific relevance and clarity in addition to more reliable and consistent predictions aligned with the study’s objectives.
Comment 3. Explain the use of generative AI in more detail.
Response 3. Thank you for the comment.
Generative artificial intelligence was used solely to enhance the wording and, to a limited extent, the syntax of textual descriptions, as well as to improve the overall fluency of the manuscript. This has been explicitly stated in Section 2.8 of the paper.
Comment 4. Which Python libraries were used?
Response 4. We thank the Reviewer for the precise suggestion, which we carefully addressed.The development and testing of the machine learning models relied primarily on the following Python libraries, listed in Section 2.8 “Data sources, code development and ethical considerations” of the manuscript:
1. pandas and numpy for data management and manipulation.
2. scikit-learn (modules: model_selection, metrics, preprocessing, ensemble) for:· data splitting (train/test)· cross-validation (e.g., cross_val_score),
performance metrics such as MAE and RMSE· preprocessing, including normalization via MinMaxScaler or StandardScaler· RandomForestRegressor.
xgboost and lightgbm for advanced boosting models (XGBRegressor, LGBMRegressor).
4. interpret (glassbox module) for the Explainable Boosting Machine (EBM), allowing interpretable models.
5. optuna for automated Bayesian hyperparameter optimization.
6. joblib for saving and loading experiments and models.
7. shap and lime for model interpretability and explainability using SHAP and LIME respectively.
8. matplotlib and seaborn for graphical visualization of results and exploratory analysis.
These libraries were used in a consistent and model-specific manner (e.g., RandomForestRegressor from sklearn, XGBRegressor from xgboost, etc.).
Comment 5. Which parameters of the machine learning methods were optimized? What was the search interval?
Response 5. We appreciate the comment, which allowed us to clarify an important part of the analysis.
For each model considered (Random Forest, XGBoost, LightGBM, and Explainable Boosting Machine - EBM), specific hyperparameters were optimized through Bayesian optimization using Optuna. The search ranges were selected to include commonly used and relevant values for each model. In detail:
• Random Forest: n_estimators (100–1000), max_depth (3–30), min_samples_split (2–10);
• XGBoost: n_estimators (100–1000), max_depth (3–10), learning_rate (0.01–0.3), subsample (0.2–1.0);
• LightGBM: n_estimators (100–1000), max_depth (-1–30), learning_rate (0.01–0.3), num_leaves (20–300), subsample (0.5–1.0), colsample_bytree (0.5–1.0);
• EBM: max_bins (64–512), max_interaction_bins (32–256), interactions (0–10), learning_rate (0.01–0.3), min_samples_leaf (2–50), max_leaves (2–64).
We have added a table in Section 3.1.3 of the manuscript that reports the parameters optimized for each model, their respective search intervals, and the optimal values found.
Comment 6. What were the parameters of the Bayesian optimization method?
Response 6. Thank you for raising this point, which helped us further strengthen the manuscript.
Bayesian optimization was carried out using the Optuna library, aiming to minimize the negative mean squared error (-MSE) estimated via cross-validation. For each model, 20 trials were performed, each with a different combination of hyperparameters, selected iteratively based on prior performance. A 3-fold cross-validation strategy was adopted to robustly evaluate model performance on the training data. All experiments were made reproducible by setting a fixed seed (random_state=1).
Comment 7. What are the optimal parameters?
Response 7. We sincerely thank the Reviewer for this useful observation, which was taken into account during the revision.
The optimal parameters for each model, identified via Bayesian optimization with Optuna and 3-fold cross validation on the training set, are as follows:
• Random Forest: n_estimators = 995, max_depth = 20, min_samples_split = 4;
• XGBoost: n_estimators = 645, max_depth = 8, learning_rate = 0.05, subsample = 0.43;
• LightGBM: n_estimators = 717, max_depth = -1, learning_rate = 0.07, num_leaves = 128, subsample = 0.91, colsample_bytree = 0.9;
• EBM: max_bins = 429, max_interaction_bins = 208, interactions = 10, learning_rate = 0.1, min_samples_leaf = 46, max_leaves = 8;
These values were selected to balance predictive performance and model complexity. These values were selected to balance predictive performance and model complexity. A summary table presenting these optimal parameters alongside their search intervals for each model has been added in Section 3.1.3 of the manuscript.
Comment 8. How many iterations with different random seeds were performed?
Response 8. Thank you for the detailed comment, which allowed us to add more clarity.
Each optimization study was carried out with 20 iterations (trials). To ensure reproducibility of results, all model training functions were executed with a fixed seed (random_state=1). This ensured consistency in hyperparameter selection and performance metrics, reducing variability due to randomness.
Comment 9. Was a K-Fold or something similar applied along with Bayesian optimization?
Response 9. We thank the Reviewer for highlighting this point, which led us to refine and clarify several steps.
During the Bayesian optimization process, model performance for each hyperparameter set was evaluated using 3-fold cross-validation. This technique allowed robust estimation of average error and helped prevent overfitting during hyperparameter tuning. The cross-validation was integrated within each Optuna trial, enabling reliable comparisons between the tested configurations.
Comment 10. Can be applied a statistical test to verify the differences are significant statistically.
Response 10. Thank you for the helpful suggestion.
As requested, a statistical analysis was performed on the distributions of the coefficient of determination (R²) on the test set. After assessing normality with the Kolmogorov–Smirnov test, Student’s t-test was applied for pairwise comparisons. The results confirmed that the differences between XGBoost and the other models (Random Forest, LightGBM, EBM) are statistically significant (p < 0.05). This information has been appropriately integrated into the text.

Reviewer 3 Report
Comments and Suggestions for Authors
This study develops an interpretable machine-learning pipeline to quantify how daily meteorological and pollutant conditions drive emergency-room admissions for cardiorespiratory diseases in metropolitan Bari, Italy. Experimental validation confirmed that every CRD admission day above the 95th percentile coincided with the simultaneous breach of these thresholds, demonstrating the rule’s reliability for early-warning systems. Therefore, I think that the paper makes a contribution and has the potential to be published. However, I summarize in the GENERAL COMMENTS as follows:
GENERAL COMMENTS
- In the introduction, the authors mention four ML models (Random Forest, XGBoost, LightGBM, and EBM) but do not explicitly justify why these specific models were chosen over others such as neural networks, SVMs.
- The study focuses on Bari, a Mediterranean city, but the Introduction does not discuss how the model might perform in regions with different climates, pollution profiles, or healthcare systems.
- While 10-fold cross-validation was used, no external validation (e.g., data from a different city or year) is provided. This limits the ability to judge whether the model generalizes beyond the Bari metropolitan area.
- While a few citations appear, this study lacks a systematic comparison with existing literature.
- In the Results section, the results are not significant enough for journal publication yet. While the study compares the ML models with traditional learning, it does not provide a detailed comparison with other existing learning models. More comprehensive evaluations are needed for journal publication.
Author Response
Comment 1. In the introduction, the authors mention four ML models (Random Forest, XGBoost, LightGBM, and EBM) but do not explicitly justify why these specific models were chosen over others such as neural networks, SVMs.
Response 1. We greatly appreciate the comment, which we found both stimulating and helpful in refining the manuscript.
As part of the study, six machine learning models were initially evaluated—Random Forest, XGBoost, LightGBM, Explainable Boosting Machine (EBM), Ridge, and TabNet—with the aim of assessing their predictive effectiveness in identifying the relationship between meteorological and environmental variables and hospital admissions for cardiorespiratory diseases. This initial selection was designed to include models that are heterogeneous in terms of complexity, interpretability, and computational approach, ranging from linear models (Ridge) to non-linear ensemble models (XGBoost, Random Forest, LightGBM), interpretable techniques (EBM), and neural networks specifically designed for tabular data (TabNet).
Preliminary results showed that Ridge and TabNet exhibited significantly lower performance compared to the other four models, both in terms of R² and test set error.
Although TabNet is a neural network, it was included specifically to explore the contribution of a deep learning architecture compatible with tabular data. However, unlike the ensemble models, it did not achieve satisfactory levels of accuracy.
Support Vector Machines (SVMs) were not included in this study because, although they can offer good performance on smaller datasets, they are computationally expensive when applied to large datasets like the one used in this study, and less flexible in handling heterogeneous numerical features without intensive preprocessing. Furthermore, their interpretability is limited compared to other algorithms employed.
In light of these findings, the final comparison focused on Random Forest, XGBoost, LightGBM, and EBM, which demonstrated not only higher predictive reliability but also the best balance between performance, interpretability, and practical applicability. This choice also allowed for a more robust and meaningful evaluation.
Comment 2. The study focuses on Bari, a Mediterranean city, but the Introduction does not discuss how the model might perform in regions with different climates, pollution profiles, or healthcare systems.
Response 2. We thank the Reviewer for the valuable observation. We have taken it into account by discussing the issue both in the Introduction and in the Discussion and Conclusion sections.
Comment 3. While 10-fold cross-validation was used, no external validation (e.g., data from a different city or year) is provided. This limits the ability to judge whether the model generalizes beyond the Bari metropolitan area.
Response 3. We thank the Reviewer for the important observation. We confirm that model validation was based on a 10-fold cross-validation procedure, which ensured robust internal validation of predictive estimates. However, we acknowledge that external validation—using data from another city or different time periods—would be an essential step to assess the true generalizability and robustness of the model in varying epidemiological and environmental contexts. Due to data availability and homogeneity constraints, such external validation was not included in the current study but is clearly indicated as a future development. An explicit mention of this aspect has been included in both the Discussion section, where limitations and future directions are discussed, and in the Conclusion, where the importance of this step for applying the model in different settings is emphasized.
Comment 4. While a few citations appear, this study lacks a systematic comparison with existing literature.
Response 4. We thank the Reviewer for the comment regarding the need for a more systematic comparison with existing literature. In response, the Introduction has been expanded to include an early reference to previous work and its limitations. Additionally, a dedicated subsection, 1.1 State of the Art, has been added, providing a detailed review of key studies using machine learning techniques to analyze the environmental impact on cardiovascular and respiratory hospital admissions. The Discussion section has also been extended to include a more explicit and systematic comparison with four selected studies of particular relevance, in order to contextualize and better highlight the value of the present findings.
Comment 5. In the Results section, the results are not significant enough for journal publication yet. While the study compares the ML models with traditional learning, it does not provide a detailed comparison with other existing learning models. More comprehensive evaluations are needed for journal publication.
Response 5. We thank the Reviewer for the observation regarding the need for a more detailed comparison with existing literature. Accordingly, an initial comparative reference was included in the Introduction and then further expanded in the new subsection 1.1 State of the Art. This section provides a focused analysis of machine learning models previously applied in similar domains, with special attention to the algorithms used and the characteristics of each study. Furthermore, a comparative reference to the literature is also present in Section 3.4.3, which anticipates more in-depth discussions presented in Section 4 Discussion. In this latter section, the comparison between our work and previous studies is developed not only in terms of the models used but also with respect to performance metrics (R² and MAE), thereby offering an objective framework that contextualizes the results within the existing scientific literature.

Reviewer 4 Report
Comments and Suggestions for Authors
In this original work, the Authors used environmental and hospitalization data recorded over 10 years to develop and validate an explainable ML framework for assessing the risk of cardiorespiratory disease (CRD) based on pollution and metereological measurements. Four ML models were developed using regionally collected data on a 10-years span and XGBoost proved to be the best performing model. SHAP and LIME analyses allowed to interpret the effective contribution of environmental variables to the outcomes. Moreover, critical thresholds of CO, atmospheric pressure, temperature and relative umidity were identified. The implications of this work may have an important impact in terms of prevention and public health. However, a few points need to be addressed to improve the clarity and robustness of the work.
Sections 3.1, 3.2 and 3.3 are better suited for the Methods section: as mainly methodological aspects are described, these (sub)sections should be moved to the Methods section.
Figure 2: it could be useful to show color legend for error bands within the figure, for better readability.
The Authors performed a 70%-30% train-test split to train the models and evaluate performances. All reported results refer to this setting: training (with cross-validation) on 70% data and testing on the remaining 30%, which is unseen during the training phase of each cv iteration. However, in the Discussion and in the Conclusions, the Authors report a phase of the study in which they validated the results, verifying the model’s ability to identify daily CRD case peak exceeding the 95th percentile threshold. Did they validate on the same data (collected from 2013 to 2023) used for model development? If so, the results may have been influenced by data leakage. Please clarify this point.
Author Response
Comment 1. In this original work, the Authors used environmental and hospitalization data recorded over 10 years to develop and validate an explainable ML framework for assessing the risk of cardiorespiratory disease (CRD) based on pollution and meteorological measurements. Four ML models were developed using regionally collected data on a 10-years span and XGBoost proved to be the best performing model. SHAP and LIME analyses allowed to interpret the effective contribution of environmental variables to the outcomes. Moreover, critical thresholds of CO, atmospheric pressure, temperature and relative humidity were identified. The implications of this work may have an important impact in terms of prevention and public health. However, a few points need to be addressed to improve the clarity and robustness of the work.
Sections 3.1, 3.2 and 3.3 are better suited for the Methods section: as mainly methodological aspects are described, these (sub)sections should be moved to the Methods section.
Response 1. We thank the Reviewer for the precise and constructive comment. We have accepted the suggestion and moved Sections 3.1, 3.2, and 3.3 from the Results section to the Materials and Methods section, adjusting the paragraph numbering accordingly. These contents are now presented as Sections 2.5, 2.6, and 2.7 within the Methodology section.
Comment 2. Figure 2: it could be useful to show color legend for error bands within the figure, for better readability.
Response 2. Thank you for drawing attention to this point, which allowed us to make a targeted improvement to the figure in question, which has been appropriately updated in the manuscript.
Comment 3. The Authors performed a 70%-30% train-test split to train the models and evaluate performances. All reported results refer to this setting: training (with cross-validation) on 70% data and testing on the remaining 30%, which is unseen during the training phase of each cv iteration. However, in the Discussion and in the Conclusions, the Authors report a phase of the study in which they validated the results, verifying the model’s ability to identify daily CRD case peak exceeding the 95th percentile threshold. Did they validate on the same data (collected from 2013 to 2023) used for model development? If so, the results may have been influenced by data leakage. Please clarify this point.
Response 3. We thank the Reviewer for the observation, as the term "validation of the results" was used incorrectly.
In fact, it was a retrospective verification of the results obtained by the model, specifically checking whether, on the days when the three thresholds were exceeded, there was simultaneously an increase in Emergency Department admissions above the 95th percentile.
Therefore, the text has been revised accordingly.

Round 2
Reviewer 1 Report
Comments and Suggestions for Authors
Thank you for revising the paper. All comments have been addressed. The paper can be accepted in the present form.
Reviewer 2 Report
Comments and Suggestions for Authors
The authors added the suggestions and answer the questions.
Reviewer 4 Report
Comments and Suggestions for Authors
The quality of the manuscript has been significantly improved and is ready for publication.